# Comparative Analysis of Flavor, Taste, and Volatile Organic Compounds in Opossum Shrimp Paste during Long-Term Natural Fermentation Using E-Nose, E-Tongue, and HS-SPME-GC-MS

**DOI:** 10.3390/foods11131938

**Published:** 2022-06-29

**Authors:** Yijia Deng, Rundong Wang, Yuhao Zhang, Xuepeng Li, Ravi Gooneratne, Jianrong Li

**Affiliations:** 1College of Food Science, Southwest University, Chongqing 400715, China; ikea7713@163.com (Y.D.); wrd2011777@163.com (R.W.); 2College of Food Science and Engineering, Bohai University, Jinzhou 121013, China; xuepengli8234@163.com; 3Chongqing Key Laboratory of Speciality Food Co-Built by Sichuan and Chongqing, Chongqing 400715, China; 4Department of Wine, Food and Molecular Biosciences, Faculty of Agriculture and Life Sciences, Lincoln University, Lincoln 7647, New Zealand; ravi.gooneratne@lincoln.ac.nz

**Keywords:** shrimp paste, flavor substances, HS-SPME-GC-MS, orthogonal partial least square discriminate analysis (OPLS-DA), correlation analysis, sensory evaluation

## Abstract

The present study focused on the determination of color, flavor, taste, and volatile organic compounds (VOCs) changes of shrimp paste fermented for 1, 2, 3, and 8 years by E-nose, E-tongue, and headspace solid-phase microextraction gas chromatography-mass spectrometry (HS-SPME-GC-MS). During fermentation, the color of shrimp paste turned dark brown with decreases in *L**, a*, and b* values. Inorganic sulfide odor was dominant in all fermented samples. The umami, richness, and aftertaste-B reached a maximum in year 3 of fermentation. A total of 182 volatiles, including long-chain alkanes, esters, aldehydes, olefins, ketones, acids, furans, and pyrazines, were detected. Sixteen VOCs including dimethyl disulfide, methional, trimethyl-pyrazine, (E,E)-2,4-heptadienal, benzeneacetaldehyde were selected as flavor markers. Correlation analysis showed that 94 VOCs were related to saltiness while 40, 17, 21, 22, and 24 VOCs contributed to richness, umami, aftertase-B, sourness, and bitterness, respectively. These novel data may help in optimizing fermentation duration to achieve target flavor indicators in opossum shrimp paste production.

## 1. Introduction

Shrimp paste is a traditional seafood seasoning used mostly in coastal areas of China and Southeast Asian countries, including Malaysia, Singapore, and Indonesia [1,2,3]. The annual production of shrimp paste in the coastal areas of Guangdong, China, reaches 10,000 tons. Shrimp paste is naturally fermented by small shrimp such as *Acete chinensis* and *Grasshopper sub* and added with 15–25% salt. During the processing stage, salt-tolerant microorganisms in shrimp paste reproduce and hydrolyze raw proteins to release peptides and amino acids by proteases to form the umami flavor [4,5], and the mixture is homogenized into a uniform paste. After long-term fermentation, we see the formation of unique volatile organic components (VOCs) such as alcohols, alkanes, aromatic hydrocarbons, esters, pyrazines, and aldehydes.

Less content of pyrazines gives shrimp paste its fresh shrimp taste, while esters and aldehydes give shrimp paste its meaty aroma and umami [6,7]. There are important indexes used to evaluate shrimp paste quality.

In shrimp paste processing, many factors such as environment temperature, salt proportion, and fermentation time significantly affect the quality and flavor of shrimp paste. Among them, the length of the fermentation period is an important factor for shrimp paste to produce the characteristic flavor. This is dependent on the complete degradation of the fat and protein by salt-tolerant bacteria in shrimp paste, especially in later stages of fermentation, which imparts the flavor. During long-term fermentation, metabolism, and succession of dominant bacterial communities such as *Marinilactibacillus* and *Pseudomonas* produce inorganic sulfides, nitrogen oxides, and along with long-chain alkanes to form the primary flavor substances of shrimp paste [8,9]. The presence of *Lactobacillus* is positively correlated with alcohols, hydrocarbons, and nitrogen-containing compounds [10,11]. All these studies showed that the length of the fermentation period was significantly correlated with the number and content of flavor compounds formed.

After investigation, we have found that most shrimp paste products sold in markets have a fermentation period of 6 months to 1 year. In order to increase the flavor and quality of shrimp paste, some food companies extend the fermentation time to 3 years. The longer fermentation time results in complete degradation of protein and hence produce more flavor substances. To date, shrimp paste research has focused mostly on the relationship between flavor substances and microorganism diversity of shrimp paste commodity [12] and isolation and characterization of halophilic lactic acid bacteria [13]. There is a lack of research on the correlation between the fermentation period and flavor. This has resulted in an inability to establish criteria based on quantifiable indicators to ensure consistent product quality and also made it difficult to screen for high-quality shrimp paste using a limited number, for example, 3 to 5 target flavor compounds. In China, many food companies that produce shrimp paste believe that a longer fermentation period contributes to a better flavor, with some extending the fermentation period up to 10 years for the premium products. However, there is no evidence to prove this. Therefore, a comparative analysis of flavor compounds formed during the fermentation period time by shrimp paste would be immensely useful in developing a screening program of target flavor indicators.

At present, the electronic nose (E-nose) and electronic tongue (E-tongue) are widely used in flavor analysis of fermented foods because they are regarded as quick, simple analytical tools for evaluating the sensory qualities of food [14,15]. Gas chromatography-mass spectrometry (GC-MS) is often used to detect volatile products during the fermentation of foods and is widely used to identify VOCs in foods such as citrus fruits and golden pompano pre-treated with simultaneous distillation extraction (SDE) [16,17]. Headspace solid-phase microextraction gas chromatography-mass spectrometry (HS-SPME-GC-MS) is an accurate method of pretreatment and analysis of fermented foods, which has been applied to accurately identify flavor compounds in Liuyang douchi [18], cider [19] and bamboo shoots [20]. The objective of this study was to comparatively analyze the composition and relative concentration of flavor volatile compounds in shrimp paste fermented for 1, 2, 3, and 8 years by HS-SPME-GC-MS, combined with changes in color, flavor, and taste measured by the color tester, E-nose, and E-tongue. Multivariate data analysis can be used to characterize and discriminate the VOCs in different shrimp pastes. These studies are of critical significance in standardizing shrimp paste quality.

## 2. Material and Methods

### 2.1. Samples Collection and Preparation

Twenty-four shrimp pastes samples (200 ± 5 g) from different ceramic fermentation containers naturally fermented for 1, 2, 3, and 8 years (6 samples were taken at each fermentation year) were collected from Bijiashan Food Co., Ltd. (Jinzhou, Liaoning province, China) and stored at 4 °C until analyses. In order to ensure the uniformity and representativeness of the shrimp paste, the samples were collected from the upper, middle, and lower parts of 10 different fermenters and combined for subsequent analysis. The shrimp paste was prepared in a Jinzhou shrimp paste-processing plant using a local and natural fermentation method. Fresh opossum shrimps were screed, cleaned, soaked, weighed, mashed, and added with 16% of salt. They were placed in ceramic fermentation containers (height 0.65 m, caliber 0.5 m, volume 80 kg), sealed with two layers of gauze, and placed outdoor for natural fermentation. The fermentation temperature is ranged from 16 to 30 °C.

### 2.2. Reagents and Chemicals

Tartaric acid (purity ≥99%), potassium chloride (≥99%), anhydrous ethanol (≥99.5%), sodium chloride (≥99%), and *n*-hexane (≥97%) were purchased from McLean Reagent Company (Shanghai, China). Benzaldehyde—D6 (CAS: 17901-93-8) purchased from Sigma-Aldrich Company (St. Louis, MO, USA) was dissolved in methanol and added to samples as an internal standard to correct for experimental error.

### 2.3. Appearance Observation and Color Analysis

The 10 g shrimp paste samples were placed on clean white paper, ‘Color Testers’ were used to detect, and the color attributes were recorded. The color attributes, including *L** (brightness), a* (greenness (−) to redness (+)), b* (blueness (−) to yellowness (+)), and △E (total color difference), were measured by CR400 Color Testers (Konica Minolia, Tokyo, Japan). The white standard tile (*L** = 99.30, a* = −0.07, b* = −0.02) was used for calibration.

### 2.4. Volatile Flavor Analysis by E-Nose

The volatile flavor analysis was detected using PEN3 Electronic Nose (Airsense Analytics Inc., Schwerin, Germany) with an auto-sampling system, sample injector, detector, data recorder, and analytical system. The 5 g shrimp paste sample and 25 mL deionized water were added to a 50 mL centrifuge tube. The mouth of the tube was wrapped with three layers of plastic sealing film and heated in a water bath at 40 °C for 20 min. The volatile gases of the sample were pumped into the sensor chamber through a needle, and the sensor signals were analyzed for 120 s. The operating temperature was 40 °C, and the sampling flow was 10 mL/min. After each sample analysis, the system was purged for 90 s with filtered air prior to the next sample injection to allow the reestablishment of the instrument baseline. The E-nose sensor array was composed of 10 metal oxide semi-conductor (MOS)-type chemical sensors. The main sensitive substances corresponding to each type of sensing elements are shown in Table 1.

### 2.5. Taste Analysis by E-Tongue

The taste analysis was conducted with an SA402B controller E-tongue (Insent Co., Ltd., Atsugi, Japan) comprising 8 chemical sensors. The chemical sensors A1 to A8 corresponded to sourness, bitterness, astringency, aftertaste-B, aftertaste-A, umami, richness, and saltiness, respectively. To 5 g shrimp paste sample in a test tube, 35 mL deionized water was added and shaken at 1700 rpm/min by a Vortex-Genie 2T (Scientific Industries, Bohemia, NY, USA) for 3 min. The solution was centrifuged at 8000 rpm/min at 4 °C for 15 min, and supernatant was collected and filtered using a 0.22 μm filter membrane. Before analysis, the sensor array was immersed in a reference solution (30 mmol/L potassium chloride and 0.3 mmol/L tartaric acid) for 30 s, then immersed in the sample solution for 120 s, and the response value at the equilibrium state was recorded for statistical analysis. After each measurement, 5% ethanol was used to clean the sensor until a stable sensor reading was obtained. Each sample was read three times in parallel, and the average was recorded for further analysis.

### 2.6. High-Throughput Analysis of Characteristic Volatile Compounds

#### 2.6.1. Sample Preparation

The 10 g shrimp paste sample (wet weight) was weighed and immediately frozen in liquid nitrogen, ground to a powder, and stored at −80 °C. The 1 g powder was added with 1 mL saturated sodium chloride solution and 10 μL benzaldehyde—D6 solution then mixed and analyzed by HS-SPME.

#### 2.6.2. Headspace Solid-Phase Microextraction (HS-SPME) Pretreatment

The SPME fiber (conditioned at 220 °C for 30 min), extraction time, extraction temperature, and sample amount were optimized based on the number of volatiles and the total peak areas. After optimization, 1 g shrimp paste powder was transferred immediately to a 20 mL headspace vial (Agilent, Palo Alto, CA, USA), and 2 mL NaCl saturated solution was added to inhibit any enzyme reaction. Vials were sealed using crimp-top caps with TFE-silicone headspace septa (Agilent Technologies, Palo Alto, CA, USA). At the time of SPME analysis, each vial was heated to 100 °C for 5 min, and 120 µm divinylbenzene/carboxen/polydimethylsilioxan fiber (Supelco, Bellefonte, PA, USA) was injected into the headspace of the sample for 15 min at 100 °C.

#### 2.6.3. Identification and Relative Quantification of VOCs by GC-MS Analysis

After sampling, desorption of the VOCs from the fiber coating was carried out in the injection port of the GC apparatus (Model 8890-5977B; Agilent) at 250 °C for 5 min in the ‘splitless’ mode. The identification and quantification of VOCs were carried out using an Agilent Model 8890 GC and a 5977B mass spectrometer (Agilent), equipped with a 30 m × 0.25 mm × 0.25 μm DB-5MS (5% phenyl-polymethylsiloxane) capillary column. Helium was used as the carrier gas at a linear velocity of 1.0 mL/ min. The injector temperature was set at 250 °C and the detector at 280 °C. The oven temperature was programmed from 40 °C (3.5 min), increasing at 10 °C/min to 100 °C, at 7 °C/min to 180 °C, at 25 °C/min to 280 °C, and held for 5 min. Mass spectra were recorded in electron impact (EI) ionization mode at 70 eV. The quadrupole mass detector, ion source, and transfer line temperatures were set at 150, 230, and 280 °C, respectively. Mass spectra were scanned in the range m/z 50–500 amu at 1 s intervals. Identification of volatile compounds was achieved by comparing the mass spectra with the data system library (NIST) and linear retention index.

### 2.7. Statistical Analysis

In all experiments, six samples of each group were analyzed, and the group results were reported as the mean ± standard deviation (SD). One-way analysis of variance (ANOVA) was used to measure significant differences with SPSS 18.0 (SPSS Inc., Chicago, IL, USA). The mean comparison was conducted using Duncan’s multiple range test [21]. Multivariate analysis (MA), including principal component analysis (PCA), orthogonal partial least square discriminate analysis (OPLS-DA), and cluster analysis (CA), were used to explore the relative variability within different varieties [22]. In high-throughput characteristic volatile compounds analysis, unsupervised PCA was performed by R software (version 3.6.1, Murray Hill, NJ, USA). The data were unit variance scaled before unsupervised PCA. The hierarchical cluster analysis (HCA) results of all volatile substances from each group were presented as heatmaps with dendrograms, while the Pearson correlation coefficient (PCC) between samples was calculated by the cor function in R and presented as heatmaps. Both HCA and PCC were carried out by an R package heatmap. For HCA, normalized signal intensities of metabolites (unit variance scaling) were visualized as a color spectrum. Significant metabolites between groups were determined by VIP ≥ 1 and absolute log_2_FC (fold change) ≥ 1. Orthogonal Partial OPLS-DA was used SIMCA-P version 13.0 software package (Umetrics, Umeå, Sweden) [23], and all variables were mean-centered and scaled to Pareto variance. VIP values were extracted from the OPLS-DA results, which also contained score plots and permutation plots, generated using the R package MetaboAnalystR. The data were log-transformed (log2) and mean-centered before OPLS-DA. In order to avoid overfitting, a permutation test (200 permutations) was performed.

## 3. Results and Discussion

### 3.1. Appearance and Color Change of Shrimp Paste

The photographic analysis of shrimp paste fermented for different time periods is shown in Figure 1. In 1-year fermented samples, shrimp paste was brown in color with an irregular granular appearance. The surface’s grainy appearance may have been due to an incomplete breakdown of proteins and fats. In longer fermented products, the color was darker and smooth with no such granular appearance. The 8-year fermented shrimp paste was dark brown. Color analysis showed that the *L** value (brightness) decreased significantly over the 1–8-year fermentation period (*p* < 0.05), with the *L** value in the 8-year samples as low as 26.77 (*p* < 0.01). The a* value gradually decreased from 1 to 3 years, but ab increased in the 8-year samples resulting in an increase in redness. The b* value remained >6 in 1- and 2-year fermented samples but significantly decreased (*p* < 0.05) in the 3- and 8-year samples. The gradual increase in △E indicates that prolonged fermentation leads to the darkening of the shrimp paste (Table 2). These results are consistent with Pongsetkul et al. (2017) [24], who reported a decrease in *L** and b* and an increased a* value, but this study was conducted only for 30 days, which is inadequate for carotenoprotein to degrade and release the carotenoids [25]. During long-term fermentation, the enzymatic browning, sometimes even blackening, is mainly caused by polyphenoloxidase (PPO) in the raw materials [26]. PPO induces hydroxylation of phenols with subsequent polymerization to form melanin [27]. Maillard reaction also contributes to the browning of fermented products [28]. Peralta et al. reported an increase in Maillard reaction products with the detection of intermediates in Philippine salt-fermented shrimp paste [29]. The brown color formed was referred to as a protein/amino acids-sugars complex formed by polymerization, condensation, degradation, and cyclization [30].

### 3.2. Volatile Flavor Composition and Identification of Taste Attributes

Volatile flavor composition in shrimp paste during 8 years of fermentation analyzed by E-nose is shown in Figure 2. Inorganic sulfides were the dominant volatile odor in year 1 but gradually declined in subsequent years. In seafood fermentation, hydrogen sulfide (H_2_S) is the main inorganic sulfide that smells such as rotten eggs, and this is possibly due to protein disruption and activity of spoilage bacteria, mostly the *Shewanella* spp. and *Serratia* spp. [31]. Therefore, H_2_S production is used as a marker to evaluate seafood spoilage [32]. In this study, a strong H_2_S smell detected in the 1-year samples significantly influenced the flavor quality of the shrimp paste. At 2–3 years of fermentation, the smell of inorganic sulfides significantly declined. Therefore, extending the fermentation period can effectively reduce the pungent H_2_S smell. In addition, nitrogen oxides (mainly NO_2_) and methyl compounds also declined during long-term fermentation. However, 8-years of fermentation resulted in a slight increase in inorganic sulfides and nitrogen oxides, which meant that shrimp paste should not be fermented for a long period. The flavor of alcohols and long-chain alkanes did not contribute to significant changes in shrimp pastes groups, probably because the flavor threshold of these is much lower compared to other flavor compounds.

The taste composition analysis of shrimp paste fermented for 1–8 years performed by E-tongue is shown in Table 3. In this study, the umami response of shrimp paste was more pronounced than other tastes and peaked in the 3-year group (*p* < 0.01). The source of umami in fermented seafood is mostly related to glycine, glutamic acid, aspartic acid, and alanine concentrations [33]. Self-degradation of protein and proteolytic enzymes caused by microorganisms results in decomposing shrimp protein into amino acids that are responsible for the umami taste. As the fermentation time increased, the bitterness (*p* < 0.05), sourness (*p* < 0.05), and astringency (*p* < 0.01) of shrimp paste decreased but the saltiness significantly increased (*p* < 0.001). According to the synergistic effect of flavor, in foods rich in umami taste, the perception of saltiness is amplified, which in turn enhances the umami taste [34]. Therefore, high-salinity shrimp paste exhibited the highest umami value in the year 3 fermentation group. The formation of astringency in shrimp paste may be due to proteolysis with the formation of small molecular peptides, resulting in an increase in hydrophobic amino acid residues, which impart a bitter taste [35]. In 8-year fermented samples, these small peptides break down to form amino acids, which reduces the astringency. The aftertaste-B values of shrimp paste samples gradually increased from 1- to 3-year samples but declined at 8 years. The aftertaste-A values also decreased during long-term fermentation. Based on PCA analysis of contribution rates of the flavor components, the first and second principal components (PC1 and PC2) were able to explain 53.44% and 32.76%, respectively of the accumulative variance contribution rate with a total contribution rate of 86.2% (Figure 3). Samples from all four-year groups, when clustered together, showed that the taste characteristics of shrimp pastes were different from each other. The taste distribution of shrimp paste fermented for years 1 and 2 were relatively close, mainly in the first quadrant, indicative of the similarity of the comprehensive taste contribution of shrimp paste during the first 2 years of fermentation. Shrimp paste fermented for 3 and 8 years were clustered in the second and third quadrants, respectively, indicating that the taste in these samples varied from the 1- and 2-year fermented groups. These data reveal that to produce a better quality shrimp paste, the fermentation duration is important. However, prolonged fermentation not only significantly reduces the umami and aftertaste of shrimp paste but also increases the contamination by harmful bacteria in the environment, including *Enterococcus* and *Streptococcus* [9,36].

### 3.3. Species and Numbers of VOCs Detected in Shrimp Paste

Through HS-SPME-GS-MS analysis, all the VOCs in shrimp paste were detected and classified. A total of 182 VOCs were identified in 24 shrimp paste samples from 4 groups. These included 39 long-chain alkanes, 22 esters, 20 aldehydes, 16 olefins, 12 ketones, 11 acids, 9 furans, 9 amines, 7 benzenoids, 5 alcohols, 5 alkanones, 5 alkynes, 4 pyrazines, 4 ethers, 3 thiophenes, 2 indoles, 2 phenols, 1 pyrene, 1 pyran, and 1 azine (Figure 4). The largest number of long-chain alkanes was mostly derived from the cleavage of long-chain fatty acid alkoxy radicals [37]. The threshold value for a long-chain alkane is not high and had little impact on shrimp paste flavor. The major contributors to the flavor of shrimp paste were esters, aldehydes, olefins, and ketones. In addition, pyrazines and indoles, which are much less, were the main contributors to the flavor.

Compared with 1- to 3-year groups, most VOCs reached the maximum in 8-year shrimp paste samples. Only a small number of VOCs, such as esters, olefins, and acids, reached a peak in the 2- and 3-year groups. In the 1-year group, the main VOCs that peaked were composed of 22 long-chain alkanes and 7 esters. In the 8-year fermented group, 14 aldehydes, 9 long-chain alkanes, 7 esters, 7 olefins, and 6 furans reached the peak (Figure 5). This shows that long-term fermentation of shrimp paste products can lead to an increase in many VOCs that affect the flavor.

### 3.4. Multivariate PCA and OPLS-DA Analyses of VOCs in Shrimp Paste

The comparison of VOCs from 1-, 2-, 3-, and 8-year fermented shrimp paste revealed the proportion of volatiles was significantly different from each other. It was difficult to distinguish the differences in flavor substances between the year groups. By performing PCA on shrimp paste samples from different fermentation years, it is possible to understand the overall flavor differences between each group and the degree of variability among samples within a group. In Figure 6A, the total variance of the VOCs in all samples was 72.01%. Based on the 3D PCA results, PC1 accounted for 39.45%, while PC2 and PC3 accounted for 21.39% and 11.17%, respectively. The score plot showed that a total of 24 samples could be separated into four characteristic groups. The data of the 1-year fermented group samples were to the left of PC1 and PC2 (negative side), with only a few samples to the right of PC3 (positive side). The 2-year fermented samples were to the left of PC1 (negative side) and to the right of PC2 and PC3 (positive side). Samples of the 3-year fermented group were mainly to the left of PC1 and PC3 (negative side) and right of PC2 (positive side). In contrast to the 1-, 2-, and 3-year sample groups, the 8-year group samples were all to the right of PC1, PC2, and PC3 (positive side). These results indicate that the amount and proportion of VOCs changed gradually during fermentation, with the compounds with a greater contribution to flavor produced mostly after 3 years of fermentation. This is supported by the CA results (Figure 6B), which show that shrimp paste fermented for 8 years had the most VOCs. The lowest VOC concentrations were in the 1-year fermentation group. The relative proportion of VOCs in 2- and 3-year groups were similar.

The pairwise between-group analysis reflected the changing patterns of VOCs at different fermentation stages. In Figure 6C, PCA indicated that VOCs from the 2-year group significantly varied from the 1-year group, with samples from the 2-year group located to the right of PC1 (57.88%, positive side). The spot of flavor distribution of the 3-year group samples was close to the 2-year group but still distinguishable with PC1 and PC2 values of 41.83% and 21.2%, respectively (Figure 6D). In the 8-year fermentation group, the flavor distribution of VOCs was significantly different from the 3-year group, principally located to the right of PC1 (58.86%, positive side), which meant an increased flavor contribution (Figure 6E).

The flavor compound data were analyzed according to the OPLS-DA model, and the scores of each group were plotted to further demonstrate the differences between the groups [38]. The prediction parameters of the evaluation model are R^2^X, R^2^Y, and Q^2^, where R^2^X and R^2^Y represent the interpretation rate of the X and Y matrices of the built model, respectively, with Q^2^ representing the predictive capability of the model. The closer these three indicators are to 1, the more stable the model is. Q^2^ > 0.9 means that it is an excellent model. Coupled with S-plot, the OPLS-DA model allows the identification of the discriminant variables, which can be further validated by the variable importance projection (VIP) plot [39]. In this study, three OPLS-DA models based on pairwise analysis of shrimp paste groups fermented for different years were established, and the results are presented in Figure 7. The R^2^X of the three models varied from 0.716 to 0.763, the R^2^Y from 0.997 to 0.998, and the Q^2^ from 0.964 to 0.989, which means that the three models had a clear separation with high-fitting quality. S-Plot figure analysis showed that some VOCs of shrimp paste samples fermented for 2 years were significantly reduced compared with the 1-year group (Figure 7(A1)). The volcano plot figure (Figure 7(A2)) shows that the number of down-regulated VOCs was 56 and the up-regulated 23. Compared with the 2-year group, 37 VOCs of the 3-year group were significantly up-regulated and 12 down-regulated (Figure 7(B2)). In the 8-year group, 55 VOCs were up-regulated and 24 down-regulated compared with the 3-year group.

Through comparative analysis, the most up-regulated VOCs in the 2-year fermentation group to be 4-acetyl-7,7-dimethyl-2-(2-oxopropyl)-cycloheptanone, 1-(2,6,6-trimethyl-1-cyclohexen-1-yl)-2-buten-1-one, benzeneacetaldehyde, 1H-indene, 2-phenylpropenal, 1H-cycloprop[e]azulene, coumarin-6-ol, methional, 1,4-benzoxazepin-3(2H)-one and pentadecane (Figure 7(A3)). In these VOCs, the ketones and aldehydes were likely derived from the oxidation and degradation of fats, which are generated on prolonged fermentation. Benzeneacetaldehyde is a fatty aldehyde derived from the decomposition of fatty acids with a sweet fruity aroma on dilution [39]. This is usually detected in fermented soybean [40], fermented black tea [41], and fermented large yellow croaker [42] and is used as a volatile marker. Similarly, benzeneacetaldehyde is also detected in shrimp paste samples [43], the production of which is linked to phenylalanine. Methional is a familiar flavor component in foods such as tomatoes [44], cheese [45], and Chinese baijiu [46]. Methional has a strong aroma of onion and meat, and the detection threshold in wine is 0.5 mg/L [47] and can contribute to food flavor. 1H-cycloprop[e]azulene detected in shrimp paste is a volatile aromatic compound detected in the essential oils of *Psidium guajava* leaves [48]. Such sesquiterpene fractions have been shown to have analgesic and anti-inflammatory properties [49]. The increased VOCs detected in the 2-year group with log_2_FC (fold change) > 1.5 and VIP values > 1.0 (compared with the 1-year group) were dimethyl tetrasulfide and dimethyl disulfide (Figure 7(A4,A5)). These two compounds were highest in the 3-year group. Kleekayai et al. have reported similar dimethyl disulfide and dimethyl trisulfide concentrations in fermented shrimp pastes [50]. The sulfur-containing flavor compound production may be generated through enzymatic hydrolysis of sulfur-containing amino acids by salt-tolerant microorganisms such as *Lactococcus lactis*, *Acetobacter pasteurianus,* and *Candida humilis* [51,52,53]. The content is small, but the taste threshold is low, which has the greater flavor contribution in fermented meat products.

The increase in the above VOCs in shrimp paste fermented for 2 years indicate that, with a prolonged fermentation period, more types of volatile substances with different structural forms can be formed, which makes the flavor more diverse. The down-regulated VOCs in 2-year group were mainly alkenes such as 1,8,11,14-heptadecatetraene, cyclooctene, bicyclo [4.2.1]nona-2,4,7-triene and 2-hexene with log_2_FC > 2.5 and VIP values > 1.3. A significant decrease in alkenes of shrimp paste may be because of the instability of the structure of carbon-carbon double bonds, which makes it easier for additive and oxidizing reactions with active groups to produce alcohols and aldehydes during fermentation. As an example, oxidation of cyclooctene can occur in the absence of catalysts [54].

In the 3-year fermentation group, the shrimp paste flavor gradually increased, and the volatile compounds up-regulated consisted mostly aldehydes, esters, and ketones. The increased aldehyde compounds were undecanal, (E,E)-2,4-heptadienal, (E,Z)-2,6-nonadienal, lilac aldehyde D, and heptanal. The ester compounds consisted mainly of cyclohexanecarboxylic acid, 3-fluorophenyl ester, oxalic acid, diallyl ester, and vinyl 10-undecenoate. The most common ketone compounds were 2,5-hexanedione, 3-amino-4,6-dimethylpyridone-2(1H) and 2,4(1H,3H)-Pyrimidinedione (Figure 7(B3)). The log_2_FC values of these flavor compounds ranged from 1.56 to 2.48 and the VIP values from 1.36 to 1.52, which means a significant increase in these compared with the 2-year group (Figure 7(B4)). During shrimp paste fermentation, an increase in aldehydes, esters, and alcohols is common, but the VOCs vary from the shrimp species or fermented method. Fan et al. reported that the flavor compounds in Chinese traditional shrimp pastes fermented by Silver shrimp, Midge shrimp, and Maxian shrimp were 3-methyl-butanal, decanal, butane(dithioic) acid, methyl ester, 2-heptanone and 2-nonanone but the concentrations varied between the shrimp species [55]. Similarly, (E,E)-2,4-heptadienal and (E,Z)-2,6-nonadienal were detected in our shrimp paste samples, which means that these two compounds can be used as volatile flavor markers. Lilac aldehyde is considered a principal olfactory molecule of lilac flowers with a distinctive aroma [56]. In our study, the lilac aldehyde concentration in shrimp paste gradually increased in the longer fermented. Lilac aldehydes are currently being developed as chemical markers of honey [57,58]. More recently, it has been identified as an odor-active compound in oysters [59]. Therefore, lilac aldehyde could also be potentially used as a flavor indicator of shrimp paste during fermentation. Four uncommon flavor VOCs increased in the 3-year group, namely 5H-5-Methyl-6,7-dihydrocyclopentapyrazine, 2-pentylfuran, N’-(2-Furoyl)-2-furohydrazide, and 2-cyclopentyl-phenol. The 5H-5-Methyl-6,7-dihydrocyclopentapyrazine is a type of flavor compound discovered in peanuts and hazelnuts with a steady nutty aroma, and at present, it is used as a new food spice produced by artificial synthesis [60]. The content of VOCs such as oxalic acid, diallyl ester, 2,4-dimethyl-quinazoline, 1-iodo-tetradecane, phytol, and cycloheptanone decreased in the 3-year group with log_2_FC values ranging from −2.68 to −3.83 and VIP values from 1.16 to 1.62 (Figure 7(B5)).

Through comparative analysis, the relative contencentration of a large number of VOCs reached a maximum in the 8-year group, which meant that the flavor complexity of shrimp paste fermented for 8 years was higher than those fermented for 3 years. The up-regulated VOCs in the 8-year group consisted mainly of 3-ethyl-2,5-dimethyl-pyrazine, heptadecane, hexadecanoic acid ethyl ester, 2,5-dibutyl-furan, para-anisaldehyde diethyl acetal, undecanal, (E,Z)-2,6-nonadienal, 2-heptadecanone, decanal (Figure 7(C3)). Among these VOCs, the log_2_FC of 3-ethyl-2,5-dimethyl-pyrazine was 5.63 and VIP value 1.24, which indicates a significant increase in the concentration of these in shrimp paste in the 8-year fermentation group (Figure 7(C4,C5)). 3-ethyl-2,5-dimethyl-pyrazine has a roasted cocoa and almond aroma, which can significantly accentuate the fermented shrimp paste flavor.

In addition, trimethyl-pyrazine was also detected in shrimp paste samples. Pyrazine is a class of aromatic substances with a distinctive nutty smell and many derivatives. The formation of pyrazines may be related to the modified Strecker reaction and Maillard reaction, which contribute to the unusual flavors and aromas in fermented products [50]. Yu et al. reported an increase in 2-methylpyrazine and 2,5-dimethyl pyrazine in shrimp paste during fermentation [43]. Based on correlation analysis, Che et al. showed that the 3-methyl-pyrazine and 2,5-dimethyl pyrazine exhibit a positive correlation with *Salimicrobium* sp., *Tetragenococcus* sp., and *Lactobacillus* sp. in shrimp paste samples during fermentation [10]. Hexadecanoic acid ethyl ester, which has a faintly waxy, fruity, and creamy aroma and is naturally found in apricot [61], jujube wine [62], and bread [63], has been used as a food additive in industrial food production. In addition to the above VOCs, compared with the 3-year group, the (E,Z)-2,6-nonadienal in shrimp paste samples was higher in the 8-year group. This volatile compound has also been detected in cucumber and has shown bactericidal activity against *Bacillus cereus*, *Escherichia coli,* and *Salmonella typhimurium* [64,65]. Therefore, the production of VOCs not only helps to increase the aroma properties of food but also prevents harmful bacterial growth. Some flavor aldehydes and alcohols, such as decanal, cuminol citral, and citronellal, also have exhibited antibacterial and antiviral activity [66,67]. The fact that shrimp paste did not spoil during 8-years of fermentation is therefore likely to be due to high salinity and the formation of volatile flavor substances with antibacterial activity at high concentrations, all of which contribute to a longer shelf life. The relative concentrations of fatty acids such as linolenic acid, lauric acid, and myristic acid also increased during the 8-year fermentation period. However, the flavor threshold of these fatty acids is relatively higher and hence not suitable as a flavor indicator [68].

### 3.5. Screening of Characteristic Flavor Compound Indexes

As shown in Table 4, based on the comparative analysis of VOCs, flavor variation, combined with taste and flavor results in all fermented shrimp samples, flavor compounds were selected as characteristic fermentation flavor indexes in shrimp paste. The flavor compounds mostly included 7 aldehydes, 2 pyrazines, 2 ketones, 2 esters, 1 furan, 1 alcohol, and 1 sulfide. The log10 peak area values of cis-2-(2-Pentenyl)furan, (E,E)-2,4-heptadienal, benzeneacetaldehyde, 3-ethyl-2,5-dimethyl-pyrazine, trimethyl-pyrazine, and 2-Tridecanone ranged from 6.02 to 6.58 at 8 years of fermentation, which is relatively higher than for the other VOCs. The relative concentration of these selected VOCs reached a peak in the 3- and 8-year fermentation groups. These selected VOCs contributed most to the flavor of shrimp paste and can in the future be used as flavor indicators in the shrimp paste production process.

### 3.6. Correlation Analysis of Taste Characteristics and Flavor Compounds

Based on the above results, umami, richness and saltiness values of shrimp pastes were significantly elevated in the long-term fermented shrimp paste samples. Through correlation analysis of taste characteristics and changes in flavor compounds in shrimp pastes were validated by Pearson correlation coefficient and significance test. The positive correlation of VOCs in shrimp paste taste are shown in Figure 8. The thicker connected lines signifies a higher correlation between taste and VOCs. In this study, we observed 94 VOCs (*p* < 0.01) showing a positive correlation with saltiness. The VOCs with a higher correlation (lower *p*-values) included N1-(4-fluorobenzyl)-N2,N2-dimethyl-1,2-ethanediamine (CAS: 2714-80-9) (*p* = 1.39 × 10^−^^11^), 2-oxo-1-methyl-3-isopropylpyrazine (CAS: 1217815-51-4) (*p* = 1.98 × 10^−10^), (Z, Z, Z)-1,8,11,14-heptadecatetraene (CAS: 10482-53-8) (*p* = 3.99 × 10^−^^10^), decanal (*p* = 4.17 × 10^−1^^0^), and 6,6-dimethyl-undecane B (CAS:17312-76-4) (*p* = 4.54 × 10^−^^10^) (Figure 8A). Otherwise, the number of VOCs related to richness, umami, aftertase-B, sourness and bitterness were 40, 17, 21, 22, and 24 (*p* < 0.01) respectively. Some VOCs showed a positive correlation with more than 2 flavors. For example, 2,3,4,9-tetrahydro-1H-pyrido [3,4-b]indol-1-one (CAS: 17952-82-8) showed a positive correlation with umami (*p* = 4.66 × 10^−^^7^), richness (*p* = 0.0002), and aftertaste-B (*p* = 0.009). The 8-methyl-5H-pyrido [4,3-b]indole (CAS: 88894-13-7) showed a positive correlation with bitterness (*p* = 0.000004), sourness (*p* = 0.00004) and astringency (*p* = 1.93 × 10^−7^). The 5-methylpyrimido [3,4-a]indole showed a significant positive correlation with bitterness (*p* = 3.68 × 10^−9^), sourness (*p* = 2.35 × 10^−5^), astringency (*p* = 8.64 × 10^−5^), richness (*p* = 0.0001), and aftertaste-B (*p* = 0.02). The screening of these flavor-related compounds is of great significance to monitor the quality of shrimp paste.

## 4. Conclusions

The sensory quality (color, flavor, and taste) and relative quantification of VOCs in shrimp paste during long-term fermentation were investigated. Correlation analysis of taste and VOCs variation was evaluated by Pearson correlation coefficient. Through comparative analysis, the values of umami, richness, and saltiness peaked after the 3-years of fermentation, while the dominant odor of inorganic sulfide in shrimp paste declined after 1-year of fermentation. Changing patterns of VOCs in shrimp paste during different years of fermentation were evident. The number of VOCs with peak concentrations was highest in the 8-year fermentation group. HS-SPME-GC-MS analysis identified a total of 182 VOCs from all shrimp paste samples, including long-chain alkanes, esters, aldehydes, olefins, ketones, acids, furans, amines, benzenoids, alcohols, alkanoic acids, alkanones, alkynes, and pyrazines. Among these VOCs, most of the long-chain alkanes reached a peak at 1-year of fermentation, while most aldehydes, pyrazines, furans, and acids reached a peak in the 8-year fermentation group. The selective flavor characteristic indexes included cis-2-(2-Pentenyl)furan, (E,E)-2,4-heptadienal, benzeneacetaldehyde, 3-ethyl-2,5-dimethyl-pyrazine, trimethyl-pyrazine, and 2-Tridecanone. In correlation analysis, 94 VOCs showed a positive correlation with saltiness, and 40, 17, 21, 22, and 24 VOCs were related to richness, umami, aftertaste-B, sourness, and bitterness, respectively. In the future, emphasis should be on screening target flavor compounds to evaluate the flavor changes during shrimp paste production.

## Figures and Tables

**Figure 1 foods-11-01938-f001:**
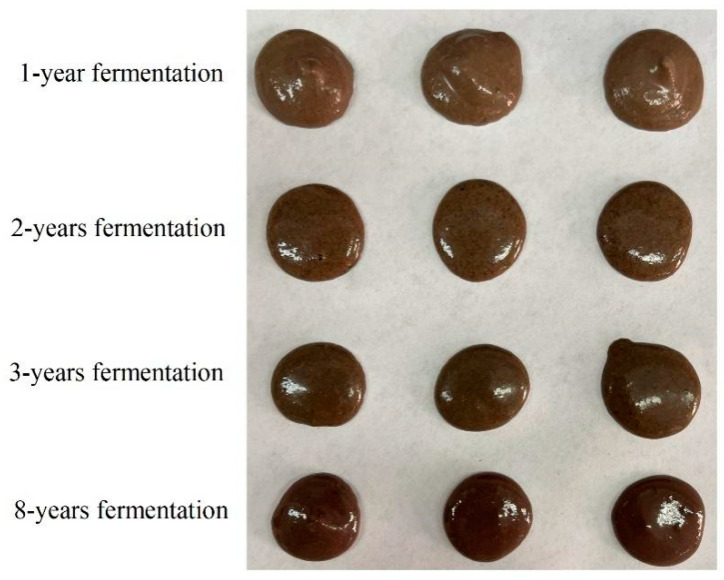
Appearance of shrimp paste at fermented for 1, 2, 3, and 8 years.

**Figure 2 foods-11-01938-f002:**
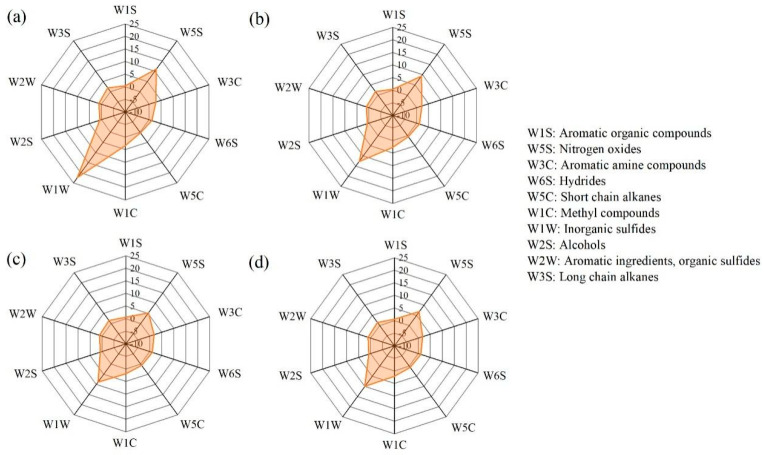
Distribution of volatile compound species in shrimp paste samples during 1 (**a**), 2 (**b**), 3 (**c**), and 8 (**d**) years of fermentation, using an E-nose sensor.

**Figure 3 foods-11-01938-f003:**
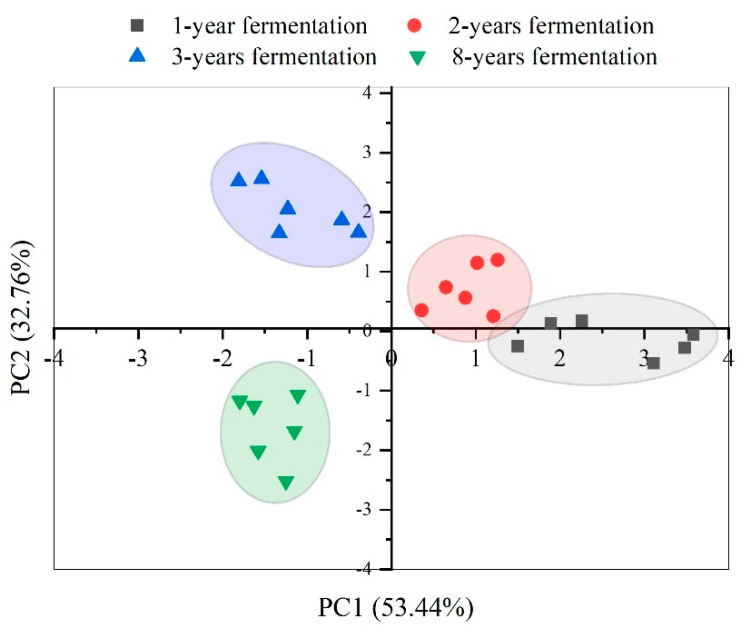
The PCA of all shrimp paste samples fermented for 1, 2, 3, and 8 years based on taste composition identified by E-tongue.

**Figure 4 foods-11-01938-f004:**
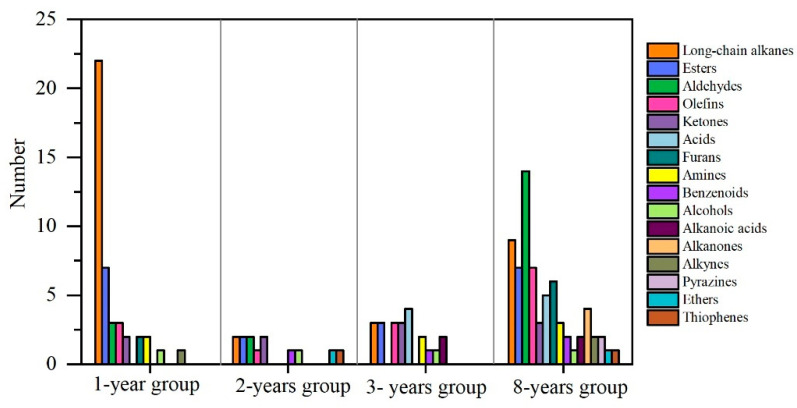
The classification and amount of VOCs in shrimp paste samples fermented for 1, 2, 3, and 8 years.

**Figure 5 foods-11-01938-f005:**
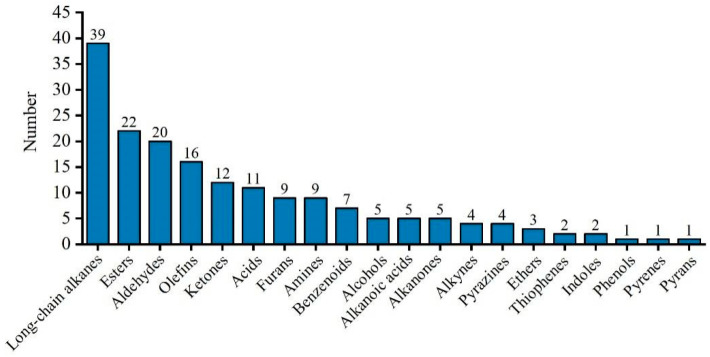
The number of peak VOCs in shrimp pastes fermented for 1, 2, 3, and 8 years.

**Figure 6 foods-11-01938-f006:**
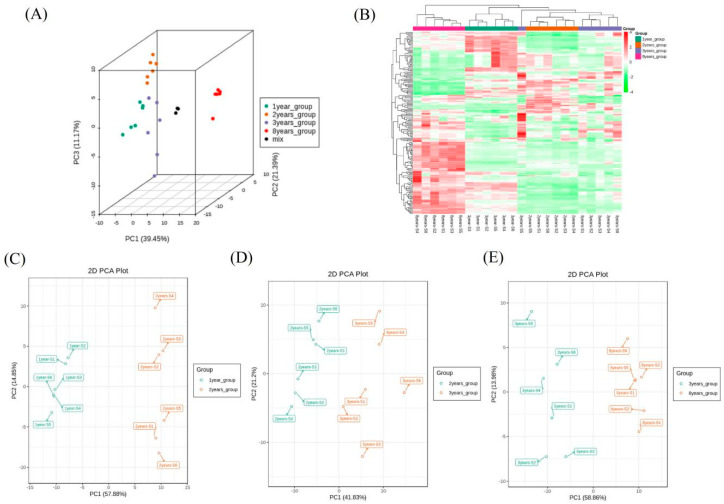
The PCA analysis of the whole groups (**A**), between two groups (**C**–**E**) and cluster analysis (**B**) of VOCs in all shrimp paste groups fermented for 1, 2, 3, and 8 years.

**Figure 7 foods-11-01938-f007:**
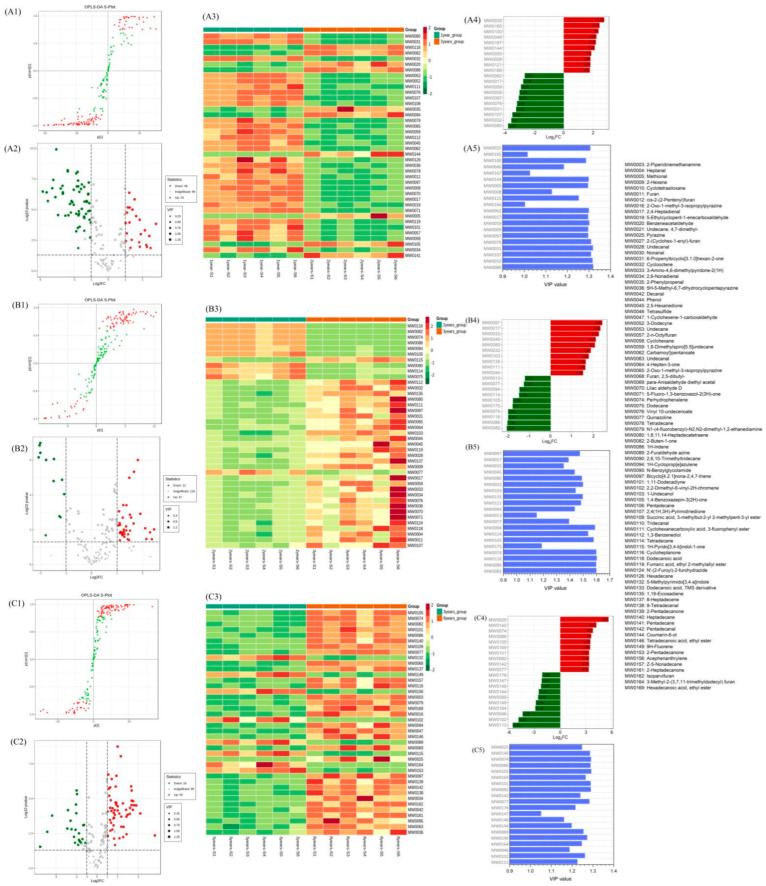
The OPLS-DA analysis (**A1**,**B1**,**C1**), between-group comparative analysis (**A2**–**A5**,**B2**–**B5**,**C2**–**C5**) of VOCs in all shrimp paste samples fermented for 1, 2, 3, and 8 years.

**Figure 8 foods-11-01938-f008:**
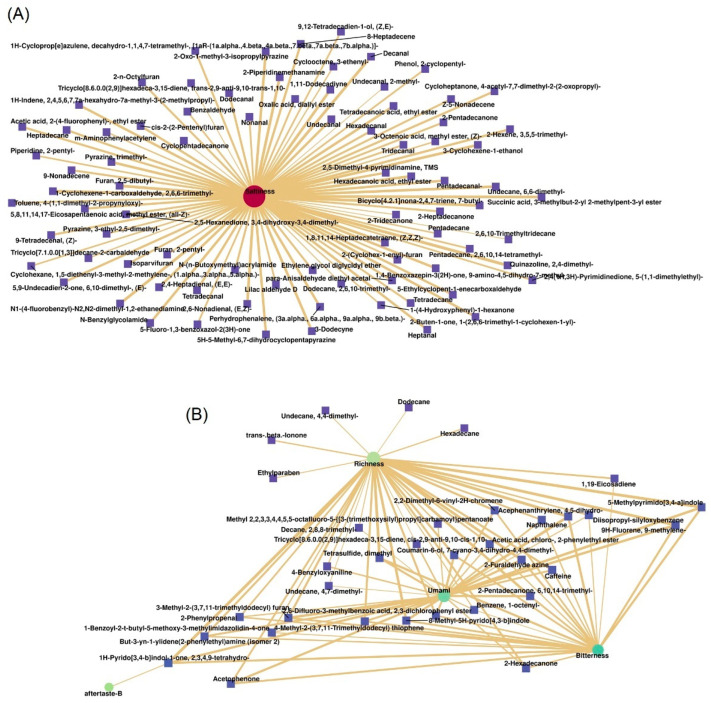
Correlation network between saltness (**A**) and richness, umami, aftertaste-B and bitterness (**B**) wiht the VOCs in shrimp paste samples fermented for 1, 2, 3, and 8 years.

**Table 1 foods-11-01938-t001:** The main compounds detected by different chemical sensors.

Sensor Number	Sensor Name	Performance Description (Sensitivity to)
R1	W1S	Aromatic compounds
R2	W5S	Nitrogen oxides, very sensitive to negative signal
R3	W3C	Aromatic amines
R4	W6S	Hydride, mainly selective for hydrogen
R5	W5C	Short-chain alkanes
R6	W1C	Methyl compounds
R7	W1W	Inorganic sulfides
R8	W2S	Alcohol
R9	W2W	Aromatic ingredients, sensitive to organic sulfides
R10	W3S	Long-chain alkanes

**Table 2 foods-11-01938-t002:** Color analyses of shrimp paste samples during 1, 2, 3, and 8 years of fermentation.

Groups (Year)	*L**	a*	b*	△E
1	36.68 ± 0.39	4.38 ± 0.16	6.28 ± 0.07	57.12 ± 0.41
2	30.27 ± 1.94 *	3.99 ± 0.12 *	6.43 ± 0.14 *	62.99 ± 1.94 *
3	29.83 ± 1.20 *	3.12 ± 0.08 **	5.85 ± 0.04 *	66.68 ± 1.24 *
8	26.77 ± 1.04 **	3.98 ± 0.24 *	5.19 ± 0.43 **	66.42 ± 1.04 *

Note: The color values of 1-year fermentation were used as the control, one-way ANOVA analysis was performed; asterisks show *p*-values < 0.05 (*), 0.01 (**) are significant.

**Table 3 foods-11-01938-t003:** Taste types of shrimp paste fermented for 1, 2, 3, and 8 years measured by E-tongue.

Flavor Types	Fermentation Duration (Years)			
	1	2	3	8
Sourness	−19.07 ± 0.12	−20.26 ± 0.35 *	−21.21 ± 0.10 *	−21.47 ± 0.23 *
Bitterness	−2.28 ± 0.16	−2.65 ± 0.08 *	−2.92 ± 0.11 *	−3.69 ± 0.07 *
Astringency	7.145 ± 0.25	6.33 ± 0.32 *	5.33 ± 0.12 **	5.21 ± 0.15 **
aftertaste-B	4.06 ± 0.06	4.52 ± 0.13 *	4.84 ± 0.10 *	3.22 ± 0.11 *
aftertaste-A	4.12 ± 0.17	3.56 ± 0.14 *	3.31 ± 0.05 *	3.3 ± 0.17 *
Umami	12.68 ± 0.26	13.25 ± 0.15 *	14.08 ± 0.10 **	13.27 ± 0.05 *
Richness	4.01 ± 0.20	5.63 ± 0.11 **	6.06 ± 0.08 **	3.61 ± 0.14 *
Saltiness	7.69 ± 0.32	8.63 ± 0.12 *	9.31 ± 0.27 **	10.06 ± 0.11 ***

Note: The taste value of the 1-year fermentation group was used as the control; one-way ANOVA analysis was performed; asterisks show *p*-values < 0.05 (*), 0.01 (**), and 0.001 (***) are significant.

**Table 4 foods-11-01938-t004:** The up-regulated VOCs in shrimp paste fermented for 1, 2, 3, and 8 years.

Index	Compounds	Retention Time (min)	CAS	Formula	Peak Area (Log10)			
					1-Year Group	2-Years Group	3-Years Group	8-Years Group
MW0001	Dimethyl disulfide	4.36	624-92-0	C_2_H_6_S_2_	5.59	5.89	6.10	5.84
MW0005	Methional	7.75	3268-49-3	C_4_H_8_OS	4.54	5.11	5.29	5.49
MW0012	Cis-2-(2-pentenyl)furan	9.48	70424-13-4	C_9_H_12_O	5.67	4.97	5.41	6.02
MW0016	Trimethyl-pyrazine	9.59	14667-55-1	C_7_H_10_N_2_	4.71	4.75	4.87	5.45
MW0017	(E,E)-2,4-heptadienal	9.75	4313-03-5	C_7_H_10_O	5.75	4.91	5.62	6.19
MW0020	Benzeneacetaldehyde	10.32	122-78-1	C_8_H_8_O	6.15	6.64	6.61	6.58
MW0025	3-Ethyl-2,5-dimethyl-pyrazine	10.91	13360-65-1	C_8_H_12_N_2_	5.05	5.24	4.76	6.25
MW0034	(E,Z)-2,6-nonadienal	12.33	557-48-2	C_9_H_14_O	5.26	5.17	5.11	5.72
MW0042	Decanal	13.31	112-31-2	C_10_H_20_O	4.39	4.42	4.66	5.03
MW0070	Lilac aldehyde	15.78	53447-47-5	C_10_H_16_O_2_	4.35	4.23	4.60	4.80
MW0081	Dodecanal	17.02	112-54-9	C_12_H_24_O	4.88	4.61	4.63	4.96
MW0103	1-Undecanol	18.47	112-42-5	C_11_H_24_O	5.17	5.05	5.59	5.52
MW0104	2-Tridecanone	18.51	593-08-8	C_13_H_26_O	5.64	5.58	5.79	6.02
MW0146	Tetradecanoic acid ethyl ester	22.64	124-06-1	C_16_H_32_O_2_	4.44	4.63	4.68	5.20
MW0161	2-Heptadecanone	23.43	2922-51-2	C_17_H_34_O	5.03	5.04	5.12	5.56
MW0169	Hexadecanoic acid ethyl ester	23.97	628-97-7	C_18_H_36_O_2_	4.02	4.29	4.49	5.20

## Data Availability

The data presented in this study are available on request from the corresponding author.

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
