# Peer review of "Comparative Analysis of Flavor, Taste, and Volatile Organic Compounds in Opossum Shrimp Paste during Long-Term Natural Fermentation Using E-Nose, E-Tongue, and HS-SPME-GC-MS"

_foods, 2022, doi:10.3390/foods11131938_

Round 1
Reviewer 1 Report
I am very grateful you for the invitation to review the manuscript foods-1765696 by Deng and co-authors "Comparative analysis of flavor, taste and volatile organic compounds in opossum shrimp paste during long-term natural fermentation using E-nose, E-tongue, and HS-SPME-GC-MS". This research analyzed the composition and relative concentration of flavour volatile compounds in shrimp paste fermented for 1, 2, 3 and 8 years by HS-SPME-GC-MS, combined with changes in color, flavour and taste measured by color tester, E-nose and E-tongue. The work is interesting but needs adjustments to increase the quality of the material.
Comments:
- Page 1, line3: Change “fermention” by “fermentation”.
- Page 1, Lines 19-20: Fermentation duration is not the only parameter. Specify that there are other factors involved in the fermentation process.
- Page 1, Lines 36-38: Change the repeated keywords by different words from the title to expand the search system.
- Page 1, Lines 41-43: Include production and consumption data in the mentioned regions.
- Pages 1-2: Lines 43-44: Shrimp paste is fermented by small shrimp such as Acete chinensis and Grasshopper or by microorganisms present in the shrimp? Describe better.
- Page 2, Line 47: characteristic flavor it's very vague. Describe better.
- Page 2, Lines 48-52: Specify how the presence of these components is interpreted for paste quality.
- Page 2, Lines 53-54: The authors need to clarify the influence of other factors such as temperature, environment, etc., on the fermentation of the paste. Only the fermentation time is not a decisive parameter for the process.
- Page 2, Lines 66-68: Similar information is presented in lines 48-52. Consider condensing the information into one place.
- Introduction: Authors must indicate the existence of legislation that establishes minimum criteria for fermented shrimp paste.
- Page 7, Figure 1: A figure with better resolution is necessary to visualize the mentioned characteristics.
- Authors must inform the criteria for choosing fermentation times. If the objective is to establish the best fermentation time for shrimp paste, a logical choice of periods would be important.
- Page 8, Lines 281-283: Based on the sentence, the fermentation time must be controlled. It is not true that longer times lead to better products. This is true up to a point. Describe better.
- Page 8, Lines 299-301: In several products, mainly dairy, the bitterness is increased with fermentation time due to protein degradation. Specify the mechanism for reducing bitterness in this case.
- Page 9, Lines 326-329: What period should be considered for the fermentation of shrimp paste, as this is one of the objectives of the study?
- Page 11, Line 355: “Most VOCs in samples peaked in the 8-year shrimp paste group.” It is important to specify that this was among the analyzed samples, but that it is impossible to say in which period this would happen, as no year-to-year follow-up was carried out.
- Page 13, Figure 6: The figure should be replaced by one of better resolution. It is impossible to view.
Page 14, Figure 7: The figure should be replaced by one of better resolution. It is impossible to view.
Page 19, Figure 8: The figure should be replaced by one of better resolution and appropriate size. It is impossible to view.
Page 20, Lines 612-614: The authors should highlight that the information is related to the datasets, as no follow-up was carried out at other times that allow us to say that this period is the maximum peak for pasta fermentation.
Author Response
Reviewer 1#
Comments:
1.- Page 1, line3: Change “fermention” by “fermentation”.
Author response: Thanks for the reviewer’s advice. We have changed “fermention” to “fermentation”.
2.- Page 1, Lines 19-20: Fermentation duration is not the only parameter. Specify that there are other factors involved in the fermentation process.
Author response: Thanks for the reviewer’s advice. This study mainly monitored the effects of different fermentation duration on the flavor of shrimp paste, but I deleted this sentence in order not to cause ambiguity.
3.- Page 1, Lines 36-38: Change the repeated keywords by different words from the title to expand the search system.
Author response:Thanks for the reviewer’s advice. We have changed the repeated keywords and changed it to another keywords to expand the search system.
4.- Page 1, Lines 41-43: Include production and consumption data in the mentioned regions.
Author response: Thanks for the reviewer’s advice. We have added the production of shrimp paste in China, but we can not find the production and consumption data in other countries. They didn't go into more detail.
5.- Pages 1-2: Lines 43-44: Shrimp paste is fermented by small shrimp such as Acete chinensis and Grasshopper or by microorganisms present in the shrimp? Describe better.
Author response: Thanks for the reviewer’s advice. We have changed this sentence. This is mainly the processing of shrimp paste, and salt-tolerant microorganisms contribute to the formation of fermented flavor, as described in the next sentence.
6.- Page 2, Line 47: characteristic flavor it's very vague. Describe better.
Author response: Thanks for the reviewer’s advice. We have changed the “characteristic flavor” to “umami flavor”.
7.- Page 2, Lines 48-52: Specify how the presence of these components is interpreted for paste quality.
Author response:Thanks for the reviewer’s advice. We have changed the sentence to specified that different flavor substance have a special significant effect on the flavour development of shrimp paste.
8.- Page 2, Lines 53-54: The authors need to clarify the influence of other factors such as temperature, environment, etc., on the fermentation of the paste. Only the fermentation time is not a decisive parameter for the process.
Author response:Thanks for the reviewer’s advice. We have added the other factors such as environment temperature and salt proportion in effect of shrimp paste’s quality and flavor. But this research that we mainly focus on the effect of fermentation time on the flavor of shrimp paste.
9.- Page 2, Lines 66-68: Similar information is presented in lines 48-52. Consider condensing the information into one place.
Author response:Thanks for the reviewer’s advice. We have deleted some repeated sentences and made the flavor information into last paragraph.
10.- Introduction: Authors must indicate the existence of legislation that establishes minimum criteria for fermented shrimp paste.
Author response: At present, the production of shrimp paste has not been standardized. It is mainly made by food companies by mashing small shrimp and fermenting them naturally. There is no uniform standard in this regard. Therefore, we explored the effect of different fermentation time on the flavor of shrimp paste, and judged the optimal fermentation time of shrimp paste by the quantity and concentration of flavor substances.
11.- Page 7, Figure 1: A figure with better resolution is necessary to visualize the mentioned characteristics.
Author response:Thanks for the reviewer’s advice. We have changed the figure with better resolution.
12.- Authors must inform the criteria for choosing fermentation times. If the objective is to establish the best fermentation time for shrimp paste, a logical choice of periods would be important.
Author response:Thanks for the reviewer’s advice. We have added the common fermentation time of shrimp paste in the third paragraph of introduction.
13.- Page 8, Lines 281-283: Based on the sentence, the fermentation time must be controlled. It is not true that longer times lead to better products. This is true up to a point. Describe better.
Author response:Thanks for the reviewer’s advice. We have changed this sentece. Longer fermentation time and significantly reduced the smell of H2S, but can not improve the flavor of shrimp paste.
14.- Page 8, Lines 299-301: In several products, mainly dairy, the bitterness is increased with fermentation time due to protein degradation. Specify the mechanism for reducing bitterness in this case.
Author response:In taste composition analysis of shrimp paste, we have discover the bitterness did not increase during 1-8 years of fermentation, so the bitterness is not the main flavor in shrimp paste. However, the astringency significantly increased in shrimp paste during fermentation and we have inferred the cause of astringency accotding to other studies.
15.- Page 9, Lines 326-329: What period should be considered for the fermentation of shrimp paste, as this is one of the objectives of the study?
Author response: At present, there is no clear standard for the fermentation time of shrimp paste, and there is no research showing that the longer the fermentation time, the better the flavor. However, some food companies believe that the longer the fermentation time, the better the flavor. Therefore, the purpose of this study is to sample and investigate from food companies, through the detection of flavor substances, to explore the change rules of flavor substances in shrimp paste with different fermentation time, and to provide a theoretical basis for the standardized production of flavored shrimp paste in the future. During fermentation, the metabolism of microorganisms is important, but we did not mentioned it. In the future, we will study the growth and metabolism of bacteria in shrimp paste during different fermentation periods.
16.- Page 11, Line 355: “Most VOCs in samples peaked in the 8-year shrimp paste group.” It is important to specify that this was among the analyzed samples, but that it is impossible to say in which period this would happen, as no year-to-year follow-up was carried out.
Author response:Thanks for the reviewer’s advice. We have changed this sentence. In this study, we only eluvated the shrimp paste fermented for 1, 2, 3 and 8 years. So compared with shrimp paste fermented for 1, 2, 3 years, most VOCs in shrimp paste fermented for 8 years reached the maximum.
- Page 13, Figure 6: The figure should be replaced by one of better resolution. It is impossible to view.
Author response:Thanks for the reviewer’s advice. We have changed the figure with better resolution.
Page 14, Figure 7: The figure should be replaced by one of better resolution. It is impossible to view.
Author response:Thanks for the reviewer’s advice. We have changed the figure with better resolution.
Page 19, Figure 8: The figure should be replaced by one of better resolution and appropriate size. It is impossible to view.
Author response:Thanks for the reviewer’s advice. We have changed the figure with better resolution.
Page 20, Lines 612-614: The authors should highlight that the information is related to the datasets, as no follow-up was carried out at other times that allow us to say that this period is the maximum peak for pasta fermentation.
Author response:Thanks for the reviewer’s advice. We have changed this sentence, the information we wanted to express is that compared with 3-year group samples, the VOCs in 8-year group was significantly higher.
Reviewer 2 Report
The work is interesting for the techniques used, the attention to the methods and correlations identified. It needs a verification of the English form in some part of the manuscript where there are errors. As regards the manuscript in general, however, some changes must be applied, as indicated below:
· in Materials the manufacturing process is not present, please add and refer the storage conditions.
· It is also necessary to specify the number of samples for each year of fermentation.
· At the same time, the shrimp paste samples must be considered different each other because the original shrimp samples I suppose are different. So the authors can discuss the results not as affect of year of fermentation, but as year of production
· Authors could explain better the experimental plan.
Considering the above comments, also the discussion of results can be modified.
Author Response
1.in Materials the manufacturing process is not present, please add and refer the storage conditions.
Author response:Thanks for the reviewer’s advice. We have added the manufacturing process of shrimp paste and the storage conditions in materials.
2.It is also necessary to specify the number of samples for each year of fermentation.
Author response:Thanks for the reviewer’s advice. We have added the number of samples for each year of fermentation.
3.At the same time, the shrimp paste samples must be considered different each other because the original shrimp samples I suppose are different. So the authors can discuss the results not as affect of year of fermentation, but as year of production
Author response: Thanks for the reviewer’s advice. For shrimp paste, the fermentation years mean how long the shrimp paste fermentation. At present, most published studies use the fermentation year to evaluate the sensory and flavor quality of shrimp paste, but they did not use production year to evaluate the quality of shrimp paste. Although the fermented shrimp paste comes from different years, the varieties and processing methods are the same. When we conduct in-depth research in the future, we will consider processing shrimp paste in the same batch, so as to conduct fermentation research and ensure the consistency of raw materials and processing methods.
4.Authors could explain better the experimental plan.
Author response:Thanks for the reviewer’s advice. We have modified some sentences in experimental part to make it more clear.
5.Considering the above comments, also the discussion of results can be modified.
Author response:Thanks for the reviewer’s advice. We have modified the discussion and results.